# Identifying Conspiracy Theories News based on Event Relation Graph

**Yuanyuan Lei and Ruihong Huang**
Department of Computer Science and Engineering
Texas A&M University, College Station, TX
{yuanyuan, huangrh}@tamu.edu

## Abstract

Conspiracy theories, as a type of misinformation, are narratives that explains an event or situation in an irrational or malicious manner. While most previous work examined conspiracy theory in social media short texts, limited attention was put on such misinformation in long news documents. In this paper, we aim to identify whether a news article contains conspiracy theories. We observe that a conspiracy story can be made up by mixing uncorrelated events together, or by presenting an unusual distribution of relations between events. Achieving a contextualized understanding of events in a story is essential for detecting conspiracy theories. Thus, we propose to incorporate an event relation graph for each article, in which events are nodes, and four common types of event relations, coreference, temporal, causal, and subevent relations, are considered as edges. Then, we integrate the event relation graph into conspiracy theory identification in two ways: an event-aware language model is developed to augment the basic language model with the knowledge of events and event relations via soft labels; further, a heterogeneous graph attention network is designed to derive a graph embedding based on hard labels. Experiments on a large benchmark dataset show that our approach based on event relation graph improves both precision and recall of conspiracy theory identification, and generalizes well for new unseen media sources[1].

## 1 Introduction

Conspiracy theories are narratives that attempt to explain the significant social or political events as being secretly plotted by malicious groups at the expense of an unwitting populations (Douglas et al., 2019; Katyal, 2002). A variety of conspiracy theories ranging from science-related moon landing to political-related pizzagate (Bleakley, 2021)

---

[1]The code and data link: https://github.com/yuanyuanlei-nlp/conspiracy_theories_emnlp_2023

are widespread throughout the world (van Prooijen and Douglas, 2018). It was estimated that more than half of the US population believed in at least one conspiracy theory (Oliver and Wood, 2014). The widespread presence of conspiracy theories can cause harm to both individuals and society as a whole (Van der Linden, 2015), such as reducing science acceptance, introducing polarization, driving violence, obstructing justice, and bringing public health risks (Hughes et al., 2022; Leonard and Philippe, 2021) etc. Thus, developing novel models to detect conspiracy theories becomes important and necessary.

Most previous work studies conspiracy theories primarily from psychology or communication perspectives, often concentrating on social media short texts (De Coninck et al., 2021; Mari et al., 2022). Few work focused on developing computational models to detect conspiracy theories, especially for lengthy news articles. However, a substantial portion of conspiracy theories originate from long documents on news websites (Miani et al., 2021), and then disseminate virally through various social media channels (Cinelli et al., 2022; Moffitt et al., 2021). Therefore, we focus on identifying conspiracy theories within these long documents from news websites, and aim to comprehend the underlying narratives. This is a challenging task which not only requires semantic understanding (Tangherlini et al., 2020) but also necessitates logical reasoning (Georgiou et al., 2021).

One observation we have for identifying conspiracy theories is that a conspiracy story can be made up by mixing unrelated events together. The concept of *"event"* refers to a specific occurrence or action that happens around us, and is essential in story telling and narrative understanding (Zhang et al., 2021). Conspiracy theories, however, usually put typically unrelated events together to fabricate a false story. Take the article in Figure 1 as an example, the author includes both *"Murder"* event

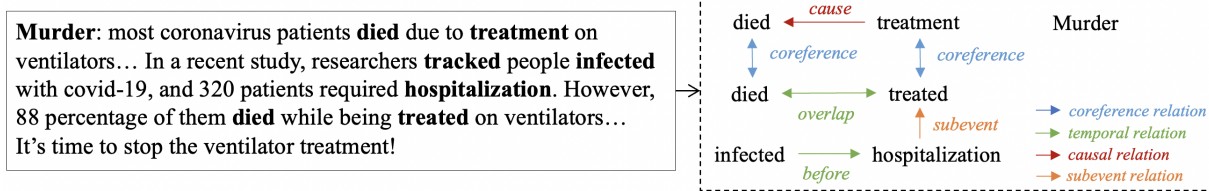

Figure 1: An example of conspiracy theory news article, and its corresponding event relation graph. Events are shown in **bold** text. Event relation graph consists of events as nodes, and four types of event-event relations as links. A conspiracy story can be made up by mixing unrelated events (nodes) together, or by presenting an irrational conspiracy logic between events (event relations).

| Conspiracy | | | | Benign | | | |
|---|---|---|---|---|---|---|---|
| Singleton | Temporal | Causal | Subevent | Singleton | Temporal | Causal | Subevent |
| **84.61%** | 30.67% | **24.18%** | **11.91%** | 76.81% | **43.25%** | 10.68% | 4.71% |

Table 1: Percentage of singleton events, temporal events, causal events, and subevents in all events reported in conspiracy theory news and benign news. The higher ratio values are shown in **bold**. Conspiracy theory articles involve less in coreference and temporal relation, but involve more in causal and subevent relation.

and *"treatment on ventilators"* event in the article, trying to persuade the readers that treatment on ventilators is equivalent to murdering, which is questionable and immediately against our intuition. To effectively guide the model to recognize the irrationality of mixing unrelated events like *"Murder"* and *"treatment on ventilators"* together, maybe by drawing upon rich world knowledge already encoded in a underlying language model, we propose to encode **events** information for conspiracy theories identification.

Furthermore, we observe that conspiracy theories often rely on presenting relations between events in an unusual way. Revisiting the example in Figure 1, the author intends to convince readers through an irrational logical chain of events: people were infected with covid-19 and required hospitalization, but died while being treated on ventilators, therefore it is the ventilator treatment that led to their death. However, the normal rational logic is that people's death just co-occurred with treatment at the same time, it was not really the treatment that caused these deaths. The conspiracy theories here presents an invalidate and baseless causal relation by emphasizing the co-occurring temporal relation. To sensitize the model to unusual and illogical event-event relations, we propose to equip the model with **event relations** knowledge. In particular, we incorporate four common types of event relations that are crucial for narrative understanding and logical reasoning (Wang et al., 2022): *coreference* - whether two event mentions designate the same occurrence, *temporal* - chronological orders

(*before*, *after* or *overlap*) between events, *causal* - causality between events, and *subevent* - containment from a parent event to a child event.

A statistical analysis of events involving the four types of event relations is shown in Table 1, based on the LOCO conspiracy theories dataset (Miani et al., 2021). The numerical analysis confirms the following observations: (1) Conspiracy theories news involves less in coreference relations, implying that unlike a normal news that reports events centered around the main event, a conspiracy theories news tends to be more dispersed in story telling. (2) Conspiracy theories exhibit less temporal relations, which means that events reported in conspiracy theories news adhere less well to the chronological narrative structure than those events presented in mainstream benign news. (3) Conspiracy theories news contains more causal relations and shows a tendency to ascribe more causality to certain events, thereby illustrating reasons or potential outcomes. (4) Conspiracy theories news employs subevent relations more frequently, elaborates more details, provides verbose explanations, and tends to incorporate more circumlocution.

Motivated by the above observations and analysis, we propose to incorporate both events and event-event relations into conspiracy theories identification. More specifically, an event relation graph is constructed for each article, in which events are nodes, and the four types of relations between events are links. This event relation graph explicitly represents content structures of a story, and guides the conspiracy theories detector to engage

its attention on significant events and event-event relations. Moreover, this event relation graph is incorporated in two ways. Firstly, an event-aware language model is trained using the soft labels derived from the event relation graph, thereby integrating the knowledge of events and event-event relations into the language model. Secondly, a heterogeneous graph attention network is designed to encode the event relation graph based on hard labels, and derive a graph feature vector embedded with both events and event relations semantics for conspiracy theories identification. Experiments on the benchmark dataset LOCO (Miani et al., 2021) show that our approach based on event relation graph improves both precision and recall of conspiracy theory identification, and generalizes well for new unseen media sources. The ablation study demonstrates the synergy between soft labels and hard labels derived from the event relation graph. Our contributions are summarized as follows:

- To the best of our knowledge, our model is the first computational model for detecting conspiracy theories news articles.

- identify events and event relations as crucial information for conspiracy theories detection.

- design a new framework to incorporate event relation graph for conspiracy theories detection.

## 2   Related Work

**Conspiracy Theory** research till now mainly studied short comments from social media such as Twitter (Wood, 2018; Phillips et al., 2022), Facebook (Smith and Graham, 2019), Reddit (Klein et al., 2019; Holur et al., 2022), or online discussions (Samory and Mitra, 2018; Min, 2021; Mari et al., 2022). However, large amount of conspiracy theories are sourced from long narratives on news websites and shared through the url link in social media platforms (Moffitt et al., 2021). Therefore, we aim to develop the model to identify conspiracy theories in these news articles.

**Misinformation Detection** was studied for years, such as rumor (Meel and Vishwakarma, 2020), propaganda (Da San Martino et al., 2019), or political bias (Fan et al., 2019; Lei et al., 2022; Baly et al., 2020; Lei and Huang, 2022). Early efforts utilized lexical analysis (Wood and Douglas, 2013). With the advent of deep learning, neural network-based models (Ma et al., 2016; Jin et al., 2017) were designed to extract semantic features (Truică and Apostol, 2022). Beyond lexical and semantic analysis, our work aims to comprehend the irrational logic behind conspiracy theories, from the perspective of events and event relations.

**Fake News Detection** is to verify whether the news article is fake or real (Pérez-Rosas et al., 2018; Rubin et al., 2015; Shu et al., 2017; Hassan et al., 2017; Thorne and Vlachos, 2018). Although both fake news and conspiracy theories involve misinformation, fake news detection focuses on checking the existence of false information, while conspiracy theories tend to explain a happened event in a malicious way through irrational logic (Avramov et al., 2020). In this paper, our attention is concentrated on exploring the intricate narratives and the underlying logic of conspiracy theories.

**Event and Event Relations** have been studied for decades. An event refers to an action or occurrence in the real world (Wang et al., 2020). The four types of event relations we consider, coreference, temporal, causal and subevent relations, are all commonly seen event-event relations in stories. However, the four types of event relations were previously studied in isolation(Zeng et al., 2020; Bethard et al., 2012; Tan et al., 2022; Yao et al., 2020). Instead, we consider all the four types of event relations to depict event-level discourse structures and achieve contextualized understanding of events in a narrative. Our approach is enabled by the recently introduced large dataset MAVEN-ERE (Wang et al., 2022) that has the four event relations annotated within individual articles.

## 3   Methodology

In this section, we explain the details of event relation graph construction for each article. The event relation graph is incorporated within two steps. Firstly, an event-aware language model is developed based on the soft labels, with events and event relations knowledge augmented. Secondly, a heterogeneous graph neural network is designed to derive a graph embedding based on the hard labels. Figure 2 illustrates our proposed methodology.

### 3.1   Event Relation Graph Construction

We need to identify events as well as extract the four types of event relations to create an event relation graph for each individual article. We will describe the training process and the graph con-

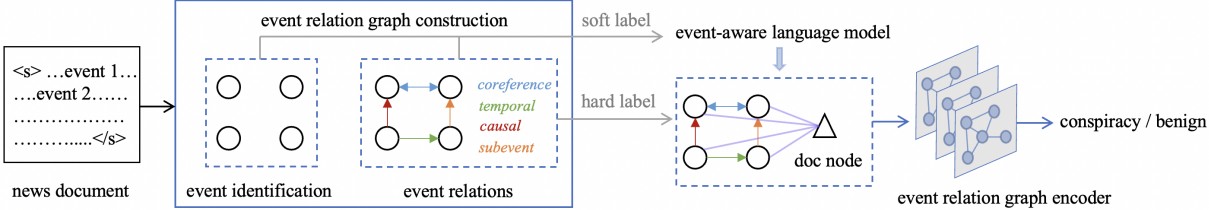

Figure 2: An illustration of conspiracy theory news identification based on event relation graph

struction process separately.

For **training**, we use the annotations from the general-domain MAVEN-ERE dataset (Wang et al., 2022). First, we train the event identifier, where the Longformer (Beltagy et al., 2020) language model is used to encode the entire article and a classification head is built on top of the word embeddings to predict whether the word triggers an event or not. Next, we train four event relation extractors. Following the previous work (Zhang et al., 2021; Yao et al., 2020), we form the training event pairs in the natural textual order, where the former event in the pair is the precedent event mentioned in text. Regarding temporal relations [2], we process the *before* annotation as follows: keep the label as *before* if the annotated event pair aligns with the natural textual order, or assign *after* label to the reverse pair if not. The *simultaneous*, *overlap*, *begins-on*, *ends-on*, *contains* annotations are grouped into the *overlap* category in our event relation graph. Regarding causal relations, we assign the *causes* label if the natural textual order is followed, and assign *caused by* label otherwise. Similarly for subevent relations, we assign the *contains* label if the natural textual order is followed, and assign *contained by* label otherwise. Since the events and four event relations interact with each other to form a cohesive narrative representation, we adopt the joint learning framework from (Wang et al., 2022) to train these components collaboratively.

During the event relation graph **construction** process, the initial step is event identification. Given a candidate news article that consists of $N$ words, the trained event identifier generates the predicted probability for each word:

$$P_i^{event} = (p_i^{non-event}, p_i^{event}) \qquad (1)$$

where $i = 1, ..., N$, and $p_i^{event}$ is the probability of $i$-th word being an event. With the events identi-

fied, the next step is to extract and classify the four types of relations. Given a pair of predicted events $(event_i, event_j)$, the trained coreference identifier predicts the probability for coreference relation as:

$$P_{i,j}^{corefer} = (p_{i,j}^{non-corefer}, p_{i,j}^{corefer}) \qquad (2)$$

where $p_{i,j}^{corefer}$ is the probability of $i$-th and $j$-th events corefer with each other. We also utilize the trained temporal classifier to generate the predicted probability for temporal relation:

$$P_{i,j}^{temp} = (p_{i,j}^{non-temp}, p_{i,j}^{before}, p_{i,j}^{after}, p_{i,j}^{overlap}) \qquad (3)$$

where $p_{i,j}^{non-temp}$ denotes the probability of no temporal relation between the $i$-th and $j$-th events, and $p_{i,j}^{before}, p_{i,j}^{after}, p_{i,j}^{overlap}$ represents the probability of *before*, *after*, *overlap* relations respectively. Similarly, the trained causal classifier and subevent classifier generate the predicted probability as:

$$P_{i,j}^{causal} = (p_{i,j}^{non-causal}, p_{i,j}^{cause}, p_{i,j}^{caused-by}) \qquad (4)$$

$$P_{i,j}^{subevent} = (p_{i,j}^{non-subevent}, p_{i,j}^{contain}, p_{i,j}^{contained-by}) \qquad (5)$$

Finally, the predicted probabilities in eq(1)-(5) are incorporated as the *soft labels* for events and event relations in the constructed event relation graph. Accordingly, the *hard label* for each element is obtained by applying the $argmax$ function to the predicted probabilities.

### 3.2 Event-aware Language Model

To incorporate the constructed event relation graph into conspiracy theories identification, we first leverage the soft labels to train an event-aware language model, with the knowledge of events and event relations augmented. The soft labels provide fuzzy probabilistic features and enable the basic language model to be aware that nodes represent events and links represent event relations.

Considering the news articles are typically long, we utilize Longformer as the basic language model

---

[2]Time expressions such as date or time are also annotated in temporal relations. However, our event relation graph focuses on events, so we exclude temporal annotations related to time expressions and solely retain annotations between events.

(Beltagy et al., 2020). In order to capture the contextual information, we add an extra layer of Bi-LSTM on top (Huang et al., 2015). Given a candidate news article, the embedding of each word is represented as $(w_1, w_2, ..., w_N)$.

To integrate **events knowledge** into the basic language model, we construct a two-layer neural network on top of the word embeddings to learn the probability of each word triggering an event:

$$
\begin{aligned}
Q_i^{event} &= (q_i^{non-event}, q_i^{event}) \\
&= softmax(W_2(W_1 w_i + b_1) + b_2)
\end{aligned}
\tag{6}
$$

where $i = 1, ..., N$, and $Q_i^{event}$ is the learned probability of the basic language model. The soft label $P_i^{event}$ generated by the event relation graph is referenced as the target learning materials. By minimizing the cross entropy loss between the learned probability $Q_i^{event}$ and the target probability $P_i^{event}$, the events knowledge from the event relation graph can be infused into the basic language model:

$$
Loss_{event} = -\sum_{i=1}^{N} P_i^{event} \log(Q_i^{event})
\tag{7}
$$

To integrate the **event relations knowledge** into the basic language model, we build four different neural networks on top of the event pair embedding, to learn the predicted probability of the four event relations respectively:

$$
\begin{aligned}
Q_{i,j}^{corefer} &= (q_{i,j}^{non-corefer}, q_{i,j}^{corefer}) \\
&= softmax(W_4(W_3(e_i \oplus e_j) + b_3) + b_4)
\end{aligned}
\tag{8}
$$

$$
\begin{aligned}
Q_{i,j}^{temp} &= (q_{i,j}^{non-temp}, q_{i,j}^{before}, q_{i,j}^{after}, q_{i,j}^{overlap}) \\
&= softmax(W_6(W_5(e_i \oplus e_j) + b_5) + b_6)
\end{aligned}
\tag{9}
$$

$$
\begin{aligned}
Q_{i,j}^{causal} &= (q_{i,j}^{non-causal}, q_{i,j}^{cause}, q_{i,j}^{caused-by}) \\
&= softmax(W_8(W_7(e_i \oplus e_j) + b_7) + b_8)
\end{aligned}
\tag{10}
$$

$$
\begin{aligned}
Q_{i,j}^{subevent} &= (q_{i,j}^{non-subevent}, q_{i,j}^{contain}, q_{i,j}^{contained-by}) \\
&= softmax(W_{10}(W_9(e_i \oplus e_j) + b_9) + b_{10})
\end{aligned}
\tag{11}
$$

where $(e_i \oplus e_j)$ is the embedding for event pair $(event_i, event_j)$ by concatenating the two events embeddings together. $Q_{i,j}^{corefer}$, $Q_{i,j}^{temp}$, $Q_{i,j}^{causal}$, and $Q_{i,j}^{subevent}$ are the learned probability of the basic language model for the four event relations. The soft labels generated by the event relation graph $P_{i,j}^{corefer}$, $P_{i,j}^{temp}$, $P_{i,j}^{causal}$, $P_{i,j}^{subevent}$ contain rich event relations information, and thus are referenced as the target learning materials. By minimizing the cross entropy loss between the learned probability and the target probability, the event relations

knowledge within the event relation graph can be augmented into the basic language model:

$$
Loss_r = -\sum_{i,j} P_{i,j}^r \log(Q_{i,j}^r)
\tag{12}
$$

where $r \in \{corefer, temp, causal, subevent\}$ represents the four event relations.

The overall loss for training the event-aware language model based on soft labels is computed as the sum of the losses for learning each component:

$$
\begin{aligned}
Loss_{soft} =\, &Loss_{event} + Loss_{corefer} + Loss_{temp} \\
&+ Loss_{causal} + Loss_{subevent}
\end{aligned}
\tag{13}
$$

### 3.3 Event Relation Graph Encoder

We further design a heterogeneous graph neural network to encode the event relation graph using hard labels. The encoder updates events embeddings with their neighbor events embeddings through interconnected relations, and produces a final graph embedding to represent each news article. This final article embedding is encoded with both events and event relations features, and will be later utilized for conspiracy theories identification.

To capture the global information of each document and explicitly indicate the connection between each document and its reported events, we introduce an extra document node and connect it to the associated events, as shown in Figure 2. The document node is initialized as the embedding of the article start token , and the event node is initialized as the event word embedding.

The resulting event relation graph comprises nine fine-grained heterogeneous relations: *coreference*, *before*, *after*, *overlap*, *causes*, *caused by*, *contains*, *contained by* relations constructed from hard labels, as well as the event-doc relation. The eight event-event relations inherently carry semantic meaning, whereas the event-doc relation is a standard link without semantics. In order to incorporate the semantic meaning of event relations into graph propagation, we introduce the relation-aware graph attention network. In terms of the event-doc relation, we utilize the standard graph attention network (Veličković et al., 2018).

The **relation-aware graph attention network** is designed to handle event-event relations, with the relations semantics integrated into event nodes embeddings. Given a pair of events $(event_i, event_j)$, their relation $r_{ij}$ is initialized as the embedding of the corresponding relation word. At the $l$-th layer, the input for $i$-th event node are output features

produced by the previous layer denoted as $h_i^{(l-1)}$. During the propagation process, the relation embedding between $i$-th and $j$-th event is updated as:

$$r_{ij} = W^r[h_i^{(l-1)} \oplus r_{ij} \oplus h_j^{(l-1)}] \qquad (14)$$

where $\oplus$ represents feature concatenation and $W^r$ is a trainable matrix. Then the attention weights $\alpha_{ij}$ across neighbor event nodes are computed as:

$$\alpha_{ij} = softmax_j\big((W^Q h_i^{(l-1)})(W^K r_{ij})^T\big) \quad (15)$$

The output features $h_{i,r}^{(l)}$ are formulated as:

$$h_{i,r}^{(l)} = \sum_{j \in \mathcal{N}_{i,r}} \alpha_{ij} W^V r_{ij} \qquad (16)$$

where $W^Q$, $W^K$, $W^V$ are trainable matrices, and $\mathcal{N}_{i,r}$ denotes the neighbor event nodes connecting with $i$-th event via the relation type $r$. After collecting $h_{i,r}^{(l)}$ for all relation types $r \in R = \{$coreference, before, after, overlap, causes, caused by, contains, contained by$\}$, we aggregate the final output feature for $i$-th event at $l$-th layer as:

$$h_i^{(l)} = \sum_{r \in R} h_{i,r}^{(l)}/|R| \qquad (17)$$

The **standard graph attention network** is employed to process event-doc relation, and update the document node embedding from connected events using the standard attention mechanism. The document node embedding at $l$-th layer is denoted as $d^{(l)}$. During propagation, the attention weights $\alpha_i$ across the connected events are computed as:

$$e_i = LeakyRelU\big(a^T[W d^{(l-1)} \oplus W h_i^{(l-1)}]\big) \qquad (18)$$

$$\alpha_i = softmax_i(e_i) = \frac{\exp(e_i)}{\sum_i \exp(e_i)} \qquad (19)$$

where $i \in$ the events set, $a$ and $W$ are trainable parameters. The final output feature for the document node at $l$-th layer is calculated as:

$$d^{(l)} = \sum_i \alpha_i W h_i^{(l-1)} \qquad (20)$$

This document node embedding is the final embedding to represent the whole article, which contains both the information of graph structure and article context. We further build a two-layer classification head on top to predict conspiracy theories, and use cross-entropy loss for training.

|  | train | dev | test |
|---|---|---|---|
| conspiracy | 39 | 11 | 8 |
| benign | 43 | 27 | 22 |

Table 2: Number of media sources in the train / dev / test set in the media source splitting setting

|  | train | dev | test |
|---|---|---|---|
| conspiracy | 4729 | 1421 | 1581 |
| benign | 14321 | 5028 | 5095 |

Table 3: Number of articles in the train / dev / test set in the media source splitting and random splitting settings

## 4 Experiments

### 4.1 Dataset

Most prior work studied conspiracy theories within social media short text. LOCO (Miani et al., 2021) is the only publicly available dataset that provides conspiracy theories labels for long news documents. LOCO consists of a large number of documents collected from 58 conspiracy theories media sources and 92 mainstream media sources. LOCO labels articles from conspiracy theories websites with the *conspiracy* class, and articles from mainstream media sources with the *benign* class. To prevent the conspiracy theory classifier from relying on stylistic features specific to a media source for identifying conspiracy stories, we will create media source aware train / dev / test data splits for our experiments as well, in addition to standard random data splitting that disregards data sources.

### 4.2 Experimental Settings

We design two different settings: 1) identifying conspiracy theories from unseen new media sources, or 2) from random media sources:

- **Media Source Splitting**: We split the dataset based on media sources, and articles from the same media source will not appear in the same set: training, development, or testing. This ensures that the model is evaluated on articles whose sources were not seen during training. This setting will remove the inflation in performance that is due to the conspiracy theory classifier relying on the shortcuts of media sources features to make prediction. Table 2 and 3 presents the statistics of media sources and articles within the train, dev, and test sets.

- **Random Splitting**: We adopt the standard

| MUC | | | $B^3$ | | | $CEAF_e$ | | | BLANC | | |
|---|---|---|---|---|---|---|---|---|---|---|---|
| Precision | Recall | F1 | Precision | Recall | F1 | Precision | Recall | F1 | Precision | Recall | F1 |
| 76.34 | 83.10 | 79.57 | 97.07 | 98.32 | 97.69 | 97.79 | 97.00 | 97.39 | 83.69 | 92.43 | 87.54 |

Table 4: Performance of event coreference resolution in the event relation graph

| | Precision | Recall | F1 |
|---|---|---|---|
| Event Identifier | 87.31 | 91.81 | 89.40 |

Table 5: Performance of event identification. Macro precision, recall, and F1 are reported.

| | Precision | Recall | F1 |
|---|---|---|---|
| Temporal | 48.45 | 46.43 | 47.04 |
| Causal | 58.48 | 54.02 | 56.01 |
| Subevent | 53.37 | 42.90 | 46.21 |

Table 6: Performance of temporal, causal, and subevent relation tasks in the event relation graph. Macro precision, recall, and F1 are reported.

method to split train / dev / test set randomly. For fair comparison, we ensure the equal size of training and development set, and also evaluate on the same testing set used in the media source splitting. In this setting, articles from the same media source can appear in both training and evaluation sets.

### 4.3 Event Relation Graph

The event relation graph is trained on MAVEN-ERE dataset (Wang et al., 2022) which contains massive general-domain news articles. The current state-of-art model framework (Wang et al., 2022) is adopted to learn different components collaboratively. The performance of event identification is presented in Table 5. Table 4 shows the results of event coreference relation identification. Following the previous work (Cai and Strube, 2010), MUC (Vilain et al., 1995), $B^3$ (Bagga and Baldwin, 1998), $CEAF_e$ (Luo, 2005), and BLANC (Recasens and Hovy, 2011) are used as evaluation metrics. Table 6 shows the performance of other components in event relation graph, including temporal, causal, and subevent relation tasks. The standard macro-average precision, recall, and F1 score are used for evaluation.

### 4.4 Baselines

Our paper presents the first attempt to identify conspiracy theories in news articles. There are few established methods available for comparison. We experimented the following systems as baselines:

- **all-conspiracy**: a naive baseline that categorizes all the documents into *conspiracy* class

- **chatgpt**: where we designed an instruction prompt (A.1) that allows the large language model ChatGPT to automatically generate predicted labels for each news article within the same test set.

- **longformer**: where we use the same language model Longformer (Beltagy et al., 2020) and add an extra layer of Bi-LSTM on top. The embedding of the article start token is used as article embedding. The same two-layer classification head is built on top of the article embedding to predict conspiracy theories. This baseline model is equivalent to our developed model without event relation graph.

- **longformer + additional features**: where we concatenate the soft labels eq (1)-(5) as additional features into the article embedding.

### 4.5 Experimental Results

Table 7 reports the experimental results of conspiracy theories news identification under two different splitting settings. Precision, recall, and F1 score of the *conspiracy* class is shown.

In the media source splitting setting, we can see that incorporating the event relation graph substantially boosts precision by 3.59% and recall by 5.13%, compared to the longformer baseline. This suggests that the event relation graph encapsulates events and their logical interrelations, thereby has the ability to understand complex narratives of conspiracy theories. These improvements also show that our proposed method can generalize well for unseen new media sources.

In the random splitting setting, incorporating the event relation graph can also bring significant improvement to both precision and recall. The results indicate the effectiveness of the event relation graph method under either the hard and easy setting, with the F1 score increased by 4.22% to 4.47%. Unsurprising, the system performance in this setting is overall higher than in the media source splitting, probably due to the inflation caused by the

| Data Split Settings | Media Source Splitting | | | Random Splitting | | |
|---|---|---|---|---|---|---|
| | Precision | Recall | F1 | Precision | Recall | F1 |
| Baseline Model | | | | | | |
| all-conspiracy | 23.68 | 100.00 | 38.30 | 23.68 | 100.00 | 38.30 |
| chatgpt | 64.77 | 28.84 | 39.91 | 64.77 | 28.84 | 39.91 |
| longformer | 79.53 | 69.32 | 74.08 | 86.59 | 75.14 | 80.46 |
| longformer + additional features | 79.37 | 70.33 | 74.58 | 86.83 | 75.90 | 80.99 |
| Event Relation Graph | | | | | | |
| + event-aware language model (soft label) | 82.30 | 70.27 | 75.81 | 89.73 | 77.92 | 83.41 |
| + event relation graph encoder (hard label) | 79.92 | 73.24 | 76.44 | 88.88 | 80.39 | 84.42 |
| + both (full model) | **83.12** | **74.45** | **78.55** | **89.22** | **80.58** | **84.68** |

Table 7: Experimental results of conspiracy theories news identification based on event relation graph under two different splitting settings. Precision, Recall, and F1 of the positive class are shown. The model with the best performance is **bold**.

classifier taking the shortcut of using media source recognition for conspiracy theory detection.

The event relation graph method outperforms the simple feature concatenation baseline. We observe that compared with the longformer baseline, incorporating probabilistic vectors as additional features (longformer + additional features) only slightly improve the performance. This demonstrate that developing more sophisticated methods to fully leverage the information embedded within the event relation graph is necessary.

The method based on event relation graph performs significantly better than the chatgpt. Comparing our full model to the chatgpt baseline, there is still a large gap in performance, especially the recall. We observe that chatgpt chooses *conspiracy* label mostly when false narratives are explicit. On the contrary, the event relation graph concentrates on logical reasoning, thus can better deal with implicit cases and yields higher recall.

### 4.6 Ablation Study

The ablation study is also shown in Table 7. The event relation graph encoder based on hard labels contributes to enhancing recall, while the event-aware language model leveraging soft labels contributes to improving precision. It is probably because that the hard labels employ the four event relations in a relatively aggressive manner, by constructing the corresponding relation regardless of the confidence level. This enables the encoder to fully exploit the intrinsic information embedded in the event relation graph, thereby enhancing recall. On the contrary, the soft labels incorporate predicted probabilities of the event relation graph, allowing the model to learn from confidence levels

| | Precision | Recall | F1 |
|---|---|---|---|
| baseline | 79.53 | 69.32 | 74.08 |
| full model | **83.12** | **74.45** | **78.55** |
| - event identify | 82.75 | 73.12 | 77.64 |
| - coreference | 82.64 | 72.87 | 77.45 |
| - temporal | 80.15 | 73.06 | 76.44 |
| - causal | 82.49 | 72.11 | 76.95 |
| - subevent | 82.11 | 72.30 | 76.89 |

Table 8: Effect of removing each of the event relation graph components: event identification, coreference, temporal, causal, and subevent relations.

and ultimately improving precision. The utilization of soft labels and hard labels are complementary with each other. Incorporating both soft labels and hard labels (the full model) exhibit the best performance.

### 4.7 Effect of Different Event Relations

We further study the effect of the different event relations in the event relation graph. Table 8 shows the experimental results of removing each type of event relations from the full model. The results show that removing any type of event relations leads to a performance drop, reducing both precision and recall. Therefore, each type of event relations plays an essential role in constructing a comprehensive content structure, and is crucial for identifying conspiracy theories news. Further, the experimental results demonstrate the importance of different event relations: removing temporal, causal, or subevent relations leads to larger performance decrease.

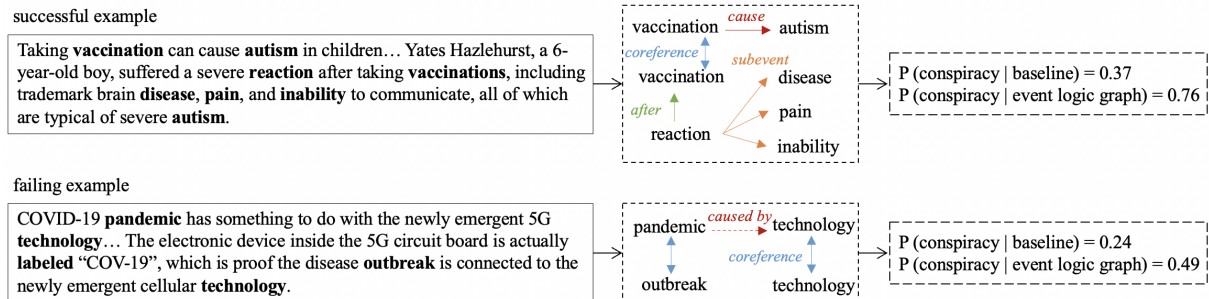

successful example

Taking **vaccination** can cause **autism** in children… Yates Hazlehurst, a 6-year-old boy, suffered a severe **reaction** after taking **vaccinations**, including trademark brain **disease**, **pain**, and **inability** to communicate, all of which are typical of severe **autism**.

failing example

COVID-19 **pandemic** has something to do with the newly emergent 5G **technology**… The electronic device inside the 5G circuit board is actually **labeled** "COV-19", which is proof the disease **outbreak** is connected to the newly emergent cellular **technology**.

Figure 3: An example of our method succeed in identifying conspiracy theory news, and a failing example. Events are shown in **bold** text. The solid arrows in the event relation graphs represent the successfully extracted event relations, and the dashed arrow means the missing event relation.

## 4.8 Analysis and Discussion

Figure 3 shows an example where our method succeeds in solving false negative. By identifying the events and extracting the relations between events, the model is aware of the author's intention to attribute a causal relation between *taking vaccination* and *autism*. At the same time, the model is encoded with the information that the *reaction* event consisting of *disease*, *pain*, and *inability* happened after *taking vaccination*, which may not be adequate to reach a causal conclusion. The model not only learns the overall distribution of the four event relations, but is also aware of the specific relations described within a story, thereby can better comprehend complex narratives.

An example where our method fails to identify conspiracy theories is also shown in Figure 3. The author intends to imply a causal relation between *COVID-19 pandemic* and *emergent 5G technology*. However, the text expresses this relation in a very implicit way by using the phrase *has something to do with*. The event relation graph algorithm fails to recognize this implicitly stated causal relation and leads to a false negative error. In order to further improve the performance of conspiracy theory identification, it is necessary to improve event relation graph construction and better extract implicit event relations.

## 5 Conclusion

This paper presents the first attempt to identify conspiracy theories in news articles. Following our observation that conspiracy theories can be fabricated by mixing uncorrelated events together or presenting an unusual distribution of relations between events, we construct an event relation graph for each article and incorporate the graph structure for conspiracy theories identification. Experimental re-

sults demonstrate the effectiveness of our approach. Future work needs to develop more sophisticated methods to improve event relation graph construction and better extract implicit event relations.

## Limitations

Our paper proposes to build an event relation graph for each article to facilitate conspiracy theories understanding. The analysis shows that the current event relation graph algorithm has the ability to extract event relations for most easy cases, but can fail in recognizing implicitly stated relations. In order to further improve the performance of conspiracy theories identification, it is necessary to develop more sophisticated methods to enhance the performance of event relation graph.

## Ethics Statement

This paper aims to identify conspiracy theories, which is a specific form of misinformation. Detecting such misinformation can promote critical thinking, mitigate misinformation, preserve trust, and safeguard societal well-being. The conspiracy theories dataset is used only for academic research purpose. The release of conspiracy theories dataset and code should only be utilized for the purpose of combating misinformation, and should not be used for spreading misinformation.

## Acknowledgements

We would like to thank the anonymous reviewers for their valuable feedback and input. We gratefully acknowledge support from National Science Foundation via the awards IIS-1942918 and IIS-2127746. Portions of this research were conducted with the advanced computing resources provided by Texas A&M High-Performance Research Computing.

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

## A Appendix

### A.1 ChatGPT Prompt

The designed instruction prompt for ChatGPT baseline is: "Conspiracy theories are narratives that explains an event or situation in an irrational or malicious manner. Please reply Yes if the following text contains conspiracy theory, else reply No. Text: xxx. Answer:"