# OpenReview forum: "Identifying Conspiracy Theories News based on Event Relation Graph"
_EMNLP/2023/Conference — EMNLP 2023 Findings_

### Official Review · Reviewer_dygx · 2023-08-02

**Soundness:** 2

**Excitement:**

3: Ambivalent: It has merits (e.g., it reports state-of-the-art results, the idea is nice), but there are key weaknesses (e.g., it describes incremental work), and it can significantly benefit from another round of revision. However, I won't object to accepting it if my co-reviewers champion it.

**Paper Topic And Main Contributions:**

In this work a model is proposed to identify conspiracy theories in news articles. The model design is based on the hypothesis that conspiracy theory identification in articles requires (or is greatly aided by) taking into account the events and their relationships as described in a text. A joint learning framework is thus adopted to extract graphs of events and their relations, which are classified as either: coreference, temporal, causal or subevent. Subsequently, the Longformer LLM (with an attention mechanism suitable for long text) is augmented with events and event relations knowledge with extra NNs to predict the classifications from the previous step, resulting in what the author(s) describe as an event-aware language model. Finally, the document token the event-aware LM and the tokens from the events are combined in a labelled graph (labelled by event relations / event-document link) that is encoded by a graph neural network, that feeds the classifier that labels the document as conspiracy theory or benign.

The event relations graph extractor is trained with the general-domain MAVEN-ERE dataset and the subsequent classifier pipeline is trained and tested with the LOCO conspiracy theory dataset. The percentages of singleton, temporal, causal and subevent cases found on the conspiracy set is found to be significantly different from the benign set, with causal and subevent connections more common in conspiracy and temporal more common in benign.

The approach presented in this work is compared against a classifier directly on top of Longformer (taking as input the article start token) and with ChatGPT prompts. The comparisons are done in two sets: by random splitting of LOCO and by media source splitting. The latter is to prevent the model from identifying artifacts of a certain publication source. In fact, both the model presented here and vanilla Longformer perform better on random split than media source split. Overall, the event relation graph approach performs much better then ChatGPT and somewhat better than vanilla Longformer on both comparisons (aprox 4.0 improvement of F-score in both cases).



**Reasons To Accept:**

This article focuses on an important contemporary topic of great public interest: the detection of conspiracy theories in news media. It uses a well justified and well designed modern NLP pipeline and it shows SOTA results in conspiracy theory detection in a human-annotated dataset of conspiracy theories and benign news.

The hypothesis that event relation knowledge graphs and reasoning about them is a crucial step in automatically identifying conspiracy theories is highly credible and worth pursuing.

**Reasons To Reject:**

I agree with the authors on the importance of reasoning about event graphs, but unfortunately I find little evidence that this is happening in this method. I find that the author(s) take a too naive (and not rigorous view) on the idea of rationality. To illustrate: in the example with the ventilator deaths, it is claimed that attributing deaths to the ventilators is an attempt to convince the reader through an "irrational chain of events" (sic). The problem is that, going by the knowledge graph contained in figure 1 alone, there is no reason to consider attribution of deaths to ventilators irrational, just because they are meant to be a treatment. It unfortunately does happen that medical treatments kill patients, and it does happen that medical practices are later found to be counter-productive etc. What makes the ventilator story into a conspiracy theory is information that is far beyond the scope of what is presented in figure 1, namely: the huge-cover up that such a thing would entail, the known deadliness of covid-19, the general opinion of experts, which in turn is based on many years of scientific publications on ventilators, respiratory diseases, etc etc. If you presented a rational person the graph of figure 1 in abstract, claiming that "there was a disease, there was a treatment, the treatment ended up being what killed the patients", this could be true. Reiterating: it is information that is not in fig. 1 that leads us to conclude that the ventilator story is unlikely to be true. Maybe the LLM encodes some general-purpose knowledge, but we don't really know what role it plays here, if any.

Then, I believe that it is telling that the F1 improvement over vanilla longformer is smaller than the difference between media and random splitting. I see no reason to assume that this model is detecting anything more than stylistic differences -- even with media split, conspiracy theories and benign news are written by different types of writers, with different types of audiences in mind, in different types of publications. I am also not convinced that this model would perform so well with completely new topics. The fact that vanilla longformer already works so well raises the question that simply a certain combination of topics is enough to detect conspiracy theories with considerable accuracy. But what about completely new topics?

I regret to say that I do not find any sufficiently significant contribution here.

**Reproducibility:**

4: Could mostly reproduce the results, but there may be some variation because of sample variance or minor variations in their interpretation of the protocol or method.

**Reviewer Confidence:**

4: Quite sure. I tried to check the important points carefully. It's unlikely, though conceivable, that I missed something that should affect my ratings.

---

> ### Author Rebuttal · Authors · 2023-08-29
>
> Thanks for your reviews and comments! Here are our answers (A) to your concerns:
>
>
> **Q1:** What makes the ventilator story into a conspiracy theory is information that is far beyond the scope of what is presented in figure 1, namely: the huge-cover up that such a thing would entail, the known deadliness of covid-19, the general opinion of experts, which in turn is based on many years of scientific publications on ventilators, respiratory diseases, etc. Reiterating: it is information that is not in fig. 1 that leads us to conclude that the ventilator story is unlikely to be true. Maybe the LLM encodes some general-purpose knowledge, but we don't really know what role it plays here.
>
>
> **A1:**
> - We completely agree with you that conspiracy theories detection is a complicated task and may rely on the general-purpose knowledge encoded within the language model. However, we want to emphasize that **the baseline model we used is also able to utilize the knowledge encoded within the same language model. The only difference between our model and baseline is the extra knowledge from event relation graph.** The non-trivial F1 improvement under both media source splitting (4.47%) and random splitting (4.22%) has demonstrated the effectiveness of incorporating the event relation graph.
> - In particular, we would like to further explain the role of our event relation graph in conspiracy theory detection. We admit that detecting the causal relation between a death event and a treatment event may not be sufficient to recognize the article is expressing a conspiracy theory, instead, **the role of event relation detection is guiding the conspiracy theory detector and engaging its attention on significant event-event relations**, and so that the conspiracy theory detector can specifically check and reason (by making use of all the knowledge its language model can provide) if it is reasonable to see “treatment on ventilators” cause deaths.
>
>
>
> **Q2:** Then, I believe that it is telling that the F1 improvement over vanilla longformer is smaller than the difference between media and random splitting. I see no reason to assume that this model is detecting anything more than stylistic differences -- even with media split, conspiracy theories and benign news are written by different types of writers, with different types of audiences in mind, in different types of publications.
>
>
> **A2:** We do not deny that stylistic features play significant roles in detecting conspiracy theories, and this is why we perform experiments in two settings. But we do not see how the relatively large performance differences between the two settings will lead to the conclusion that our model is only detecting stylistic differences.
> - It is worthy to note that **the performance gains in the random splitting setting are solely from the features of the event relation graph.** Within the random splitting setting where both our model and baseline can leverage the media source stylistic features, incorporating the event relation graph features can improve F1 by 4.22% (80.46% -> 84.68%). This indicates that event relation graph features provide extra complementary information to the stylistic features.
> - **The event relation graph features can also improve performance when the stylistic features are not available or limited.** In order to remove or alleviate the impact of media source style features, we also design the media source splitting setting, where both our model and baseline can not use the media source stylistic features. The results show that incorporating event relation graph can also increase F1 by 4.47%, which proves the usefulness of the event relation graph when the stylistic features are limited.
>
>
> In summary, the performance gains of our proposed model over baseline under both settings demonstrate the effectiveness of the event relation graph, whether the stylistic features are available or not.
>
>
>
>
> **Q3:** I am also not convinced that this model would perform so well with completely new topics . The fact that vanilla longformer already works so well raises the question that simply a certain combination of topics is enough to detect conspiracy theories with considerable accuracy. But what about completely new topics?
>
>
> **A3:**
> - **Our experiments demonstrate that the conspiracy theories detector relies more on the media source features than the topic features.** We did an experiment in the topic-based splitting setting, where the model detects conspiracy theories in unseen new topics. The experimental results are shown in the table below. Compared to the random splitting, mitigating the influence of media source features (media source splitting) leads to bigger performance drop than mitigating the influence of topic features (topic splitting). This phenomenon demonstrates that the conspiracy theories detector relies more on the media source features than the topic features. Therefore, we primarily focus on media source splitting in the paper due to the space limit.
> - **Our experiments indicate the effectiveness of event relation graph, whether media source features or topic features are available or not.** Comparing our proposed model (Row 4) with the baseline (Row 1), the performance gains indicate that incorporating event relation graph can increase precision and recall under media source splitting, topic splitting, and random splitting settings, regardless of the availability of media source features or topic features. We also did the ablation study of the two designed components under the topic-based splitting setting, as shown in the table (Row 2-3). We will add the results into the paper.
>
>
>
> | Data Split Settings | Media Source Splitting | Topic Splitting | Random Splitting |
> | ------------------- | ---------------------- | --------------- | ---------------- |
> | | Precision / Recall / F1 | Precision / Recall / F1 | Precision / Recall / F1 |
> | longformer (baseline) | 79.53 / 69.32 / 74.08 | 85.75 / 75.01 / 80.02 | 86.59 / 75.14 / 80.46 |
> | + event aware language model (soft label) | 82.30 / 70.27 / 75.81 | 88.58 / 78.36 / 83.16 | 89.73 / 77.92 / 83.41 |
> | + event relation graph encoder (hard label) | 79.92 / 73.24 / 76.44 | 86.17 / 81.40 / 83.72 | 88.88 / 80.39 / 84.42 |
> | + both (event relation graph full model) | 83.12 / 74.45 / 78.55 | 88.01 / 81.72 / 84.75 | 89.22 / 80.58 / 84.68 |
>
>
>
> **Q4:** I regret to say that I do not find any sufficiently significant contribution here.
>
> **A4:** Our primary **contributions** are summarized as follows for reference:
>
> * We are **the first paper to detect conspiracy theories news**. Although conspiracy theories are wide-spread and harmful, there are no developed computational models for detecting such misinformation yet, especially in long news documents.
> * We **find out events and event relations are crucial for conspiracy theories detection.** Our empirical observations and statistical analysis collectively affirm that conspiracy theories news can present an unusual connection between events -- departing from the typical *temporal* narrative structure but frequently relying on *causal* relations and *subevent* relations to convince the readers.
> * We **propose and build the event relation graph** to integrate four common event relations together. Furthermore, we also study the interactions and combination effect of the four relations, which were not well studied previously. We also design a new framework to incorporate the event relation graph into conspiracy theories detection.

---

### Official Review · Reviewer_9T2j · 2023-08-04

**Soundness:** 3

**Excitement:**

3: Ambivalent: It has merits (e.g., it reports state-of-the-art results, the idea is nice), but there are key weaknesses (e.g., it describes incremental work), and it can significantly benefit from another round of revision. However, I won't object to accepting it if my co-reviewers champion it.

**Paper Topic And Main Contributions:**

This paper studied the problem of detecting conspiracy theories, a type of misinformation that explains an event or situation in an irrational or malicious manner, in a news article. The authors proposed to construct a event relation graph for each article and incorporate the graph structure to identify conspiracy theories. Experimental results show good results of the proposed method.

**Questions For The Authors:**

This paper studied an interesting problem on detecting conspiracy and provided some useful insights into the problem. I have a few questions.

1. The authors highlighted that most previous studies on conspiracy detection have focused on social media short texts. It would be useful to explain how previous methods on short texts cannot work well on long news articles. This also relates to the choice of baselines in the experiments. As the dataset was released recently, there might not be available baselines that have been directly applied to the dataset. I think it would be useful to include some state-of-the-art baselines that were used for short text in this case --- to show the superiority of the proposed method but also show the differences between short and long text domains. The current baselines look weak: one is a "reduced" version of the proposed method, one is a general unsupervised language model, and another one is even not a genuine classifier.

2. I found a bit hard to follow the section 3.2 on event-ware language model and I am not sure why this step is needed. I guess that this step is to incorporate both contextual information in a raw document and event relations extracted from section 3.1. It would be useful to make it clearer.

3. Although the experimental results show that the hard labels are useful, I feel that hard labels are redundant, as they are a transferred version of soft labels using the argmax function. In fact, the soft labels might provide rich information due to their fuzzy/probability nature, particularly when there is ambiguity (or overlapping between multiple relations) between a pair of events. Just a suggestion. I would explore whether it is possible to use a vector <P_event, P_corefer, P_temp, ..... ,> as weights or representation of an edge between events, and drop hard labels to make the whole process neater.

4. In Figure 3, events that have a relation of coreference often share the same string. Should we collapse these same strings as a single node to reduce the size of graph and training time?

**Reasons To Accept:**

Interesting research question; Rich details; promising results.

**Reasons To Reject:**

Baselines were slightly weak; some steps are unclear;

**Reproducibility:**

4: Could mostly reproduce the results, but there may be some variation because of sample variance or minor variations in their interpretation of the protocol or method.

**Reviewer Confidence:**

4: Quite sure. I tried to check the important points carefully. It's unlikely, though conceivable, that I missed something that should affect my ratings.

**Typos Grammar Style And Presentation Improvements:**

"articles from the same media source will not appear in the same set" -- > "articles from the same media source will appear in the same set"  in "Media Source Splitting" P6.

"89.73" is higher than "89.22" and should be bold in Table 4.

---

> ### Author Rebuttal · Authors · 2023-08-29
>
> Thanks for your valuable suggestions and questions! Here are our answers (A) to your questions (Q):
>
>
> **Q1:** Baselines were slightly weak. The authors highlighted that most previous studies on conspiracy detection have focused on social media short texts. I think it would be useful to include some state-of-the-art baselines that were used for short text in this case  --- to show the superiority of the proposed method but also show the differences between short and long text domains.
>
> **A1:** We do not present previous work on social media short text as baselines because most of previous work discuss psychology or communication perspectives of conspiracy theories from social media, instead of developing computational models to detect them.
> - **Most previous work studied conspiracy theories from psychology or communication perspectives, and few work focused on computational detection model building.** The work such as *De Coninck et al. 2021*, *Min Seong Jae 2021*, *Mari Silvia et al. 2022* studied the problem of who believes in conspiracy theories, from psychology perspective. The work such as *Wood 2018*, *Smith and Graham 2019*, *Klein et al 2019* researched the dissemination pattern of conspiracy theories on social media platforms, from a communication perspective. To the best of our knowledge, *Phillips et al. 2022* is the only work that engaged with computational model to detect conspiracy theories in social media short text.
> - To alleviate your concern, we build an additional baseline based on the method released in *Phillips et al. 2022*, and the results demonstrate that **the model developed for social media short text is not capable to detect conspiracy theories in long news articles.** We show the results under two experimental settings in the first row of the table below. There exists a noticeable performance gap between the model (Phillips et al. 2022) developed for social media short text and our longformer baseline developed for news. This indicates the different characteristics between social media short text and news articles. Understanding lengthy news articles requires profound comprehension towards news structure and complicated narratives.
>
>
> | Data Split Settings | Media Source Splitting | Random Splitting |
> | ------------------- | ---------------------- | ---------------- |
> |     | Precision / Recall / F1    | Precision / Recall / F1 |
> | *Phillips et al. 2022 (baseline)* | 57.12 / 48.88 / 52.67 | 66.01 / 53.62 / 59.17 |
> | longformer (baseline) | 79.53 / 69.32 / 74.08 | 86.59 / 75.14 / 80.46 |
> | event relation graph (our method) | 83.12 / 74.45 / 78.55 | 89.22 / 80.58 / 84.68 |
>
>
> >[1] De Coninck, David, et al. "Beliefs in conspiracy theories and misinformation about COVID-19: Comparative perspectives on the role of anxiety, depression and exposure to and trust in information sources." Frontiers in psychology 12 (2021): 646394.\
> >[2] Min, Seong Jae. "Who believes in conspiracy theories? Network diversity, political discussion, and conservative conspiracy theories on social media." American Politics Research 49.5 (2021): 415-427.\
> >[3] Mari, Silvia, et al. "Conspiracy theories and institutional trust: examining the role of uncertainty avoidance and active social media use." Political Psychology 43.2 (2022): 277-296.\
> >[4] Wood, Michael J. "Propagating and debunking conspiracy theories on Twitter during the 2015–2016 Zika virus outbreak." Cyberpsychology, behavior, and social networking 21.8 (2018): 485-490.\
> >[5] Smith, Naomi, and Tim Graham. "Mapping the anti-vaccination movement on Facebook." Information, Communication & Society 22.9 (2019): 1310-1327.\
> >[6] Klein, Colin, Peter Clutton, and Adam G. Dunn. "Pathways to conspiracy: The social and linguistic precursors of involvement in Reddit’s conspiracy theory forum." PloS one 14.11 (2019): e0225098.\
> >[7] Holur, Pavan, et al. "Which side are you on? Insider-Outsider classification in conspiracy-theoretic social media." In Proceedings of the 60th Annual Meeting of the Association for Computational Linguistics (Volume 1: Long Papers), pages 4975–4987, Dublin, Ireland. Association for Computational Linguistics.\
> >[8] Phillips, Samantha C., Lynnette Hui Xian Ng, and Kathleen M. Carley. "Hoaxes and hidden agendas: A twitter conspiracy theory dataset" Companion Proceedings of the Web Conference 2022. 2022.
>
>
>
>
> **Q2:** Just a suggestion. I would explore whether it is possible to use a vector <P_event, P_corefer, P_temp, ..... ,> as weights or representation of an edge between events.
>
>
> **A2:** Thanks for your suggestion. We follow your suggestion to add another baseline of concatenating the soft label vector <P_event, P_corefer, P_temp, P_causal, P_subevent> as additional features into the article embedding. The results under two experimental settings are reported in the second row of the table below. We observe that, compared with the longformer baseline, incorporating soft labels as additional features only slightly improve the performance. One possible explanation is that simple feature concatenation cannot fully exploit the information embedded within the event relation graph. **The results demonstrate that our event relation graph approach still performs better than the baseline with probability features.**
>
>
> | Data Split Settings | Media Source Splitting | Random Splitting |
> | ------------------- | ---------------------- | ---------------- |
> |     | Precision / Recall / F1    | Precision / Recall / F1 |
> | longformer (baseline) | 79.53 / 69.32 / 74.08 | 86.59 / 75.14 / 80.46 |
> | *longformer + additional features* | 79.37 / 70.33 / 74.58 | 86.83 / 75.90 / 80.99 |
> | event relation graph (our method) | 83.12 / 74.45 / 78.55 | 89.22 / 80.58 / 84.68 |
>
>
>
>
> **Q3:** I found a bit hard to follow the section 3.2 on event-ware language model and I am not sure why this step is needed. I guess that this step is to incorporate both contextual information in a raw document and event relations extracted from section 3.1. It would be useful to make it clearer.
>
>
> **A3:** **The purpose of event-aware language model is to distill the events and event relations knowledge from soft labels into the fundamental language model.** We build neural networks on top of the fundamental language model, guiding it to learn from soft labels. Therefore, the events and event relations knowledge within the soft labels can be distilled into the fundamental language model, resulting in an event-aware language model. Without this event-aware language model, the fundamental language model does not know that nodes in the graph are events and links are event relations.
>
>
>
>
>
> **Q4:** Although the experimental results show that the hard labels are useful, I feel that hard labels are redundant, as they are a transferred version of soft labels using the argmax function. In fact, the soft labels might provide rich information due to their fuzzy/probability nature, particularly when there is ambiguity (or overlapping between multiple relations) between a pair of events.
>
>
> **A4:** The hard labels are essential in our proposed method. We would like to make several points about the benefits of incorporating hard labels that can not be achieved by incorporating soft labels:
> - **The hard labels enable us to represent the event relation graph as a graph structure.** The event-aware language model based on the soft labels has events and event relations knowledge, but does not have the graph structural information. The hard labels enable us to explicitly link the event nodes through four relations, and eventually form a graph structure.
> - **The hard labels enable us to update event embeddings with their neighbor events information.** The graph structure constructed based on the hard labels enables us to update the event embeddings with their neighbor events information through connected relations. (section 3.3 relation-aware graph attention network)
> - **The hard labels enable us to derive a final graph embedding for conspiracy theories detection.** The graph structure facilitates us to derive a final graph embedding to represent the document, which is further used for conspiracy theories detection, while soft labels can not. (section 3.3 standard graph attention network)
>
>
>
>
> **Q5:** In Figure 3, events that have a relation of coreference often share the same string. Should we collapse these same strings as a single node to reduce the size of graph and training time?
>
>
> **A5:** **The two events with the same trigger word can have different context**, such as event arguments and other contextual information, **and thus should have different embeddings**. Therefore, we kept a node for each event mention in the graph.

---

### Official Review · Reviewer_nzA4 · 2023-08-05

**Typos Grammar Style And Presentation Improvements:** 1- "through the url" -> through the U…
**Soundness:** 4

**Excitement:**

4: Strong: This paper deepens the understanding of some phenomenon or lowers the barriers to an existing research direction.

**Missing References:**

In addition to random splitting, using source-based splitting is great. I can suggest referring to studies like the ones below in this line:
1- Hürriyetoğlu, A., Yörük, E., Mutlu, O., Duruşan, F., Yoltar, Ç., Yüret, D., & Gürel, B. (2021). Cross-context news corpus for protest event-related knowledge base construction. Data Intelligence, 3(2), 308-335.

2- Hürriyetoğlu, A., Yörük, E., Yüret, D., Yoltar, Ç., Gürel, B., Duruşan, F., & Mutlu, O. (2019). A task set proposal for automatic protest information collection across multiple countries. In Advances in Information Retrieval: 41st European Conference on IR Research, ECIR 2019, Cologne, Germany, April 14–18, 2019, Proceedings, Part II 41 (pp. 316-323). Springer International Publishing.

**Paper Topic And Main Contributions:**

The authors propose a method for identifying conspiracy theories that are based on an event graph constructed using events as nodes and event relations such as coreference, causality, temporal, and subevent as relations. This task is important and has not been studied much on long texts.  The method proposed in this report is compared to strong baselines. Moreover, the cross-source evaluation, in addition to random splitting, provides a realistic evaluation. The error analysis and ablation study enable a reader to understand both the task and merits of the proposed methodology for conspiracy detection in long text such as text.

**Questions For The Authors:**

What is the training time and the parameters you used or evaluated to reach the results reported in the paper?



**Reasons To Accept:**

1- The task is important.

2- The task and the method are described well.

3- The evaluation is realistic and shows that the proposed system achieves good performance.

4- The paper is written well.

**Reasons To Reject:**

The reproducibility of the paper could be strengthened by providing dependencies/requirements of the code submitted with this report.

**Reproducibility:**

2: Would be hard pressed to reproduce the results. The contribution depends on data that are simply not available outside the author's institution or consortium; not enough details are provided.

**Reviewer Confidence:**

3: Pretty sure, but there's a chance I missed something. Although I have a good feel for this area in general, I did not carefully check the paper's details, e.g., the math, experimental design, or novelty.

---

> ### Author Rebuttal · Authors · 2023-08-29
>
> Thanks for your valuable comments! Here are our answers (A) to your questions (Q):
>
> **Q1:** The reproducibility of the paper could be strengthened by providing dependencies/requirements of the code submitted with this report.
>
> **A1:** We will provide detailed instructions for the released version of code.
>
> **Q2:** What is the training time and the parameters you used or evaluated to reach the results reported in the paper?
>
> **A2:** The training time of our model is around 8 hours on one GPU of Nvidia RTX 3090. The AdamW (Loshchilov and Hutter, 2019) is used as the optimizer. The learning rate is initialized as 1e-5 and then adjusted by a linear scheduler. The weight decay is set to be 1e-2. The maximum length of input is set to 4096. The number of training epochs is 5. We will add these information into the paper.
>
> **Q3:** Missing References and Typos
>
> **A3:** We will add the references you suggested into the paper, and correct the typos.

---

### Meta-Review · Area_Chair_vaCJ · 2023-09-09

**Recommendation:** 3

**Metareview:**

This work proposes a computational model to detect what the authors describe as conspiracy theories in news articles. The authors hypothesize that “conspiracy theories” will contain what they describe as “uncorrelated events” or an “unusual distribution of relations between events.”  Based on this hypothesis, they propose a model which attempts to use both the events in a news article (e.g. an infected event or hospitalization event) and the relationships between such events (e.g. infected before hospitalization) to detect conspiracy theories in news articles. The authors evaluate their model on an existing dataset LOCO (Miani et al., 2021), which appears to use credible news sources and conspiracy news sources to label individual news documents. The authors also emphasize that their work is different from prior studies, which focus on detecting conspiracies in shorter social media posts.

Overall, the reviewers offered a mixed assessment of this work.

Reviewer nzA4 offered a shorter review emphasizing the importance of the task, the quality of the exposition and the soundness of the experiments. This reviewer also praised the authors for testing their model with source-based splitting (i.e., the model had to predict conspiracies in unseen news sources), which seems important for checking that the model can actually learn to detect substantive markers of conspiracy theories instead of just stylistic features of conspiracy news sources. This reviewer gave the paper a soundness score of 4 and an excitement score of 4.

Reviewer 9T2j also believed the experiments were sound (awarding a score of 3). But they were less excited about thew work than reviewer nzA4, awarding an excitement score of 3. In their review, 9T2j asked for additional evidence that models designed for shorter texts would perform less well on longer news articles. In response, the authors added an additional baseline in the rebuttal period. They showed that this method had F1 scores that were nearly 20 points lower than models designed for longer text. Reviewer 9T2j also asked about if it was necessary to include hard labels in the model, and the authors discussed their role in their rebuttal. This discussion appears less relevant because the authors compared these approaches empirically in Table 4. Reviewer 9T2j did not respond to the rebuttal.

Reviewer dygx offered lower scores, and asked questions about the nature of conspiracy theories in their review. They hypothesized about the kinds of reasoning that might lead people to believe conspiracy theories, and the extent to which LLMs might already have representations of such reasoning. These comments from dygx were just a hypothesis without grounding in experiments or prior work, and should not be given too much weight. But the idea of a “conspiracy theory” could have been much more sharply defined in the submission. The authors do seem to show that extracting events and event relations can detect a conspiracy news source in the LOCO dataset. But the work could have been richer and more convincing if the authors used their model to explore the kinds of event relations which may mark conspiracy theories, and the ways in which these patterns can inform our understanding of what constitutes a conspiracy theory. For instance, the authors might provide an exploratory analysis of the kinds of spurious event relations that are common to conspiracy theories.

Reviewer dygx also asked an interesting question surrounding soundness. They hypothesized that there might be high-level stylistic differences between conspiracy and non-conspiracy sources which might undermine the findings from the experiments in this paper. If all conspiracy and non-conspiracy sources share stylistic similarities (e.g., if each source is drawn hierarchically from a parent conspiracy or non-conspiracy distribution) then source-based splitting might not be sufficient to check that models are not just learning broad stylistic differences between kinds of news sources. However, in their rebuttal, the authors noted that including event information improved F1 scores in the random split setting (i.e., naively splitting the dataset without considering source). In this setting, the authors argue, event and baseline models each had access to the same stylistic information. But including event relations lead to a 4.22 point increase in F1 score over a vanilla longformer model (80.46 => 84.68). Reviewer dygx did not respond to this argument.

Finally, because of these concerns about stylistic information, the reviewer dygx also noted that they are “not convinced that this model would perform so well with completely new topics.” The authors did attempt to address this comment from dygx in their rebuttal. But their experiments on “topic splitting” were hard to assess because they were lacking detail. The authors do not appear to describe “topic features” in their paper so it is hard to understand this new experiment.

---

### Decision · Program_Chairs · 2023-10-07

**Decision:**

Accept-Findings

**Comment:**

This work proposes a computational model to detect what the authors describe as conspiracy theories in news articles. The authors hypothesize that “conspiracy theories” will contain what they describe as “uncorrelated events” or an “unusual distribution of relations between events.”  Based on this hypothesis, they propose a model which attempts to use both the events in a news article (e.g. an infected event or hospitalization event) and the relationships between such events (e.g. infected before hospitalization) to detect conspiracy theories in news articles. The authors evaluate their model on an existing dataset LOCO (Miani et al., 2021), which appears to use credible news sources and conspiracy news sources to label individual news documents. The authors also emphasize that their work is different from prior studies, which focus on detecting conspiracies in shorter social media posts.

Overall, the reviewers offered a mixed assessment of this work.

Reviewer nzA4 offered a shorter review emphasizing the importance of the task, the quality of the exposition and the soundness of the experiments. This reviewer also praised the authors for testing their model with source-based splitting (i.e., the model had to predict conspiracies in unseen news sources), which seems important for checking that the model can actually learn to detect substantive markers of conspiracy theories instead of just stylistic features of conspiracy news sources. This reviewer gave the paper a soundness score of 4 and an excitement score of 4.

Reviewer 9T2j also believed the experiments were sound (awarding a score of 3). But they were less excited about thew work than reviewer nzA4, awarding an excitement score of 3. In their review, 9T2j asked for additional evidence that models designed for shorter texts would perform less well on longer news articles. In response, the authors added an additional baseline in the rebuttal period. They showed that this method had F1 scores that were nearly 20 points lower than models designed for longer text. Reviewer 9T2j also asked about if it was necessary to include hard labels in the model, and the authors discussed their role in their rebuttal. This discussion appears less relevant because the authors compared these approaches empirically in Table 4. Reviewer 9T2j did not respond to the rebuttal.

Reviewer dygx offered lower scores, and asked questions about the nature of conspiracy theories in their review. They hypothesized about the kinds of reasoning that might lead people to believe conspiracy theories, and the extent to which LLMs might already have representations of such reasoning. These comments from dygx were just a hypothesis without grounding in experiments or prior work, and should not be given too much weight. But the idea of a “conspiracy theory” could have been much more sharply defined in the submission. The authors do seem to show that extracting events and event relations can detect a conspiracy news source in the LOCO dataset. But the work could have been richer and more convincing if the authors used their model to explore the kinds of event relations which may mark conspiracy theories, and the ways in which these patterns can inform our understanding of what constitutes a conspiracy theory. For instance, the authors might provide an exploratory analysis of the kinds of spurious event relations that are common to conspiracy theories.

Reviewer dygx also asked an interesting question surrounding soundness. They hypothesized that there might be high-level stylistic differences between conspiracy and non-conspiracy sources which might undermine the findings from the experiments in this paper. If all conspiracy and non-conspiracy sources share stylistic similarities (e.g., if each source is drawn hierarchically from a parent conspiracy or non-conspiracy distribution) then source-based splitting might not be sufficient to check that models are not just learning broad stylistic differences between kinds of news sources. However, in their rebuttal, the authors noted that including event information improved F1 scores in the random split setting (i.e., naively splitting the dataset without considering source). In this setting, the authors argue, event and baseline models each had access to the same stylistic information. But including event relations lead to a 4.22 point increase in F1 score over a vanilla longformer model (80.46 => 84.68). Reviewer dygx did not respond to this argument.

Finally, because of these concerns about stylistic information, the reviewer dygx also noted that they are “not convinced that this model would perform so well with completely new topics.” The authors did attempt to address this comment from dygx in their rebuttal. But their experiments on “topic splitting” were hard to assess because they were lacking detail. The authors do not appear to describe “topic features” in their paper so it is hard to understand this new experiment.